# Characteristics and Survival Outcomes of Hepatocellular Carcinoma Developed after HCV SVR

**DOI:** 10.3390/cancers13143455

**Published:** 2021-07-09

**Authors:** Ming-Lun Yeh, Po-Cheng Liang, Pei-Chien Tsai, Shu-Chi Wang, Jennifer Leong, Eiichi Ogawa, Dae Won Jun, Cheng-Hao Tseng, Charles Landis, Yasuhito Tanaka, Chung-Feng Huang, Jun Hayashi, Yao-Chun Hsu, Jee-Fu Huang, Chia-Yen Dai, Wan-Long Chuang, Mindie H. Nguyen, Ming-Lung Yu

**Affiliations:** 1Department of Internal Medicine and Hepatitis Center, Hepatobiliary Division, Kaohsiung Medical University Hospital, Kaohsiung 807, Taiwan; 930291@kmuh.org.tw (M.-L.Y.); 970396@kmuh.org.tw (P.-C.L.); 1090102@kmuh.org.tw (P.-C.T.); 910063@kmuh.org.tw (C.-F.H.); jfliver@kmu.edu.tw (J.-F.H.); 820195@kmuh.org.tw (C.-Y.D.); waloch@kmu.edu.tw (W.-L.C.); 2Lipid Science and Aging Research Center, School of Medicine, and Hepatitis Research Center, College of Medicine, and Center for Cancer Research and Center for Liquid Biopsy and Cohort Research, Kaohsiung Medical University, Kaohsiung 807, Taiwan; 3Department of Medical Laboratory Science and Biotechnology, Kaohsiung Medical University, Kaohsiung 807, Taiwan; shuchiwang@kmu.edu.tw; 4Henry D. Janowitz Division of Gastroenterology, Mt. Sinai Health System, New York, NY 10029, USA; jennifer.leong@mountsinai.org; 5Department of General Internal Medicine, Kyushu University Hospital, Fukuoka 812-8582, Japan; eogawa@gim.med.kyushu-u.ac.jp; 6Department of Gastroenterology, School of Medicine, Hanyang University, Seoul 04763, Korea; noshin@hanyang.ac.kr; 7Division of Gastroenterology of Hepatology, E-Da Hospital, I-Shou University, Kaohsiung 82445, Taiwan; ed104817@edah.org.tw (C.-H.T.); ed111996@edah.org.tw (Y.-C.H.); 8Department of Gastroenterology, University of Washington Medical Center, Seattle, WA 98195, USA; clandis@medicine.washington.edu; 9Department of Gastroenterology and Hepatology, Faculty of Life Sciences, Kumamoto University, Kumamoto 860-8555, Japan; ytanaka@med.nagoya-cu.ac.jp; 10Kyushu General Internal Medicine Center, Haradoi Hospital, Fukuoka 812-0063, Japan; hayashij1949@haradoi-hospital.com; 11Division of Gastroenterology and Hepatology, Department of Medicine, Stanford University Medical Center, Palo Alto, CA 94304, USA; 12Department of Epidemiology and Population Health, Stanford University, Palo Alto, CA 94304, USA

**Keywords:** hepatitis C virus, chronic hepatitis C, hepatocellular carcinoma, viremia, survival

## Abstract

**Simple Summary:**

It is important to understand the impact of viremia on the survival of hepatocellular carcinoma (HCC) in hepatitis C patients. This study aimed to investigate the characteristics and survival between hepatitis C patients with and without viremia at HCC diagnosis. We enrolled 1,389 HCC patients, including 301 with HCC developed after hepatitis C eradication (post-SVR HCC) and 1,088 with hepatitis C viremia (viremic HCC). Post-SVR HCC patients had better liver function, earlier tumor stages and higher median survival than viremic HCC patients. But post-SVR HCC was not independently associated with survival on further multivariate analysis. On sub-analysis, viremic HCC patients who subsequently eradicated hepatitis C had higher median survival and was also significantly associated with lower mortality as compared to post-SVR HCC. Therefore, the advantages in clinical and tumor characters determined the better overall survival of post-SVR HCC patients; however, eradication of hepatitis C after HCC also improved survival.

**Abstract:**

The clinical presentation and survival of hepatocellular carcinoma (HCC) after hepatitis C virus (HCV) eradication as compared to HCC in viremic patients are not well characterized. We aimed to investigate the characteristics and survival between HCV patients with and without viremia at HCC diagnosis.: We retrospectively analyzed overall survival outcomes in 1389 HCV-related HCC patients, including 301 with HCC developed after HCV eradication (post-SVR HCC) and 1088 with HCV viremia at HCC diagnosis (viremic HCC). We also evaluated overall survival in the two groups using propensity score-matching methods.: At HCC diagnosis, post-SVR HCC patients were older, less obese, less likely cirrhotic, with better liver function, lower alfa-fetoprotein levels, earlier BCLC stages, and higher rate of treatment with surgery. Overall, post-SVR HCC patients had higher median survival than viremic patients (153.3 vs. 55.6 months, *p* < 0.01), but post-SVR HCC was not independently associated with survival on multivariate analysis (adjusted HR: 1.05, 95% CI: 0.76–1.47). However, on sub-analysis, viremic HCC patients who subsequently received anti-viral treatment and achieved SVR had higher median survival than post-SVR HCC patients (*p* < 0.01). Viremic HCC with subsequent SVR was also significantly associated with lower mortality as compared to post-SVR HCC (adjusted HR: 0.18, 95% CI: 0.11–0.29). In addition, we observed similar findings in our analysis of the propensity score-matched cohorts.: The advantages in clinical and tumor characters at HCC diagnosis determined the better overall survival of post-SVR HCC patients; however, HCV eradication after HCC development was also associated with improved survival.

## 1. Introduction

Hepatitis C virus (HCV) infection is a major cause of hepatocellular carcinoma (HCC) globally [1,2]. Sustained virological response (SVR, undetectable HCV RNA PCR 12–24 weeks after completion of therapy) has been shown to substantially decrease the risk of HCC development though not complete risk elimination [3,4]. More recently, the introduction of well-tolerated and highly efficacious direct acting antivirals (DAA) even among those with established HCC has allowed many more HCV-related HCC (HCV-HCC) patients to be treated, as these patients generally have advanced liver disease and comorbidities that disqualify them for interferon (IFN)-based treatment [5,6,7,8]. Unfortunately, there is still a large proportion of viremic HCV patients who remain undiagnosed for HCV and/or untreated even in DAA era [9,10].

Recent data have also shown that HCV-HCC patients treated with DAA have higher survival than untreated patients [11], but it is not clear if there are differences in patient and tumor characteristics as well as survival outcomes of patients who developed HCC while viremic versus those who developed HCC after SVR by DAA or IFN-based therapies. Since patient and tumor characteristics at HCC diagnosis are the major determinants of long-term clinical outcomes, it is important to characterize these factors in these two patient populations [12,13]. Therefore, we aimed to evaluate a large and diverse cohort of HCV-HCC patients to characterize and compare the clinical presentation and long-term survival of HCV-HCC patients by the presence of HCV viremia at HCC diagnosis. In addition, for those who were still viremic at the time of HCC diagnosis, we evaluated and compared the subgroup who subsequently underwent antiviral therapy and achieved SVR versus those who remained viremic.

## 2. Materials and Methods

This retrospective study analyzed data collected at nine clinical centers from four countries or regions (three in the United States, two in Japan, one in Korea, and three in Taiwan). Patients were enrolled in this study if they fulfilled all the following criteria: (1) Having a diagnosis of chronic hepatitis C defined by a positive anti-HCV and/or HCV RNA for more than six months; (2) having a diagnosis of HCC as determined by histology/cytology or by typical imaging findings by contrast-enhanced computerized tomography or magnetic resonance imaging [14,15]; (3) having HCV RNA and treatment completion data to determine viremic and SVR status in relation to HCC diagnosis time. We excluded patients with hepatitis B virus co-infection or HCC diagnosis within 6 months of achieving HCV SVR (to avoid the potential confounding of a pre-existed HCC). Enrolled patients were categorized as viremic HCC (patients of positive HCV RNA at HCC diagnosis) and post-SVR HCC (patients of negative HCV RNA at HCC diagnosis). Patients of viremic HCC were further divided into a viremic HCC with the subsequent SVR group (patients who achieved HCV SVR after HCC diagnosis) and viremic HCC who remained the viremic group (patients who were untreated or failed to achieve HCV SVR after HCC diagnosis). The study allocation flow chart is shown in Figure 1.

The study was conducted in accordance with the ethical principles of the Declaration of Helsinki in 1975, as revised in 2008, and was approved by the institutional review board of each participating institution.

### 2.1. Baseline Characteristic Evaluation

Data of clinical and tumor characteristics were collected via review of medical records at each participating institution using a unified data frame and data variable definition. Laboratory data including platelet count, aspartate aminotransferase (AST), alanine aminotransferase (ALT), total bilirubin, albumin, prothrombin time (INR, international normal ratio), creatinine, and alpha-fetoprotein (AFP) levels were collected. HCV RNA was measured by qualitative or quantitative polymerase chain reaction assays as available at the time of examination. HCC characteristics, including tumor number (single or multiple), size of largest tumor, macroscopic vessel invasion, extra-hepatic metastasis, Barcelona Clinic Liver Cancer (BCLC) stage [14], and initial primary HCC treatment were obtained. The primary HCC treatment was categorized as curative therapy (e.g., surgery, local ablation with curative intent, and liver transplantation), palliative therapy (e.g., trans-arterial chemoembolization, external radiotherapy, and systemic therapy), and supportive care.

For the 143 patients who received surgical resection, pathological characteristics of resected tissues were also recorded. Metabolic disorder was defined with the presence of either obesity, hypertension, and/or diabetes. Liver cirrhosis was determined by imaging, laboratory, or clinical evidence of portal hypertension such as splenomegaly, nodular liver, thrombocytopenia, ascites, and encephalopathy. Impaired liver status was defined as Child-Pugh score ≥7.

### 2.2. Outcome Evaluation

The primary outcome was overall survival which was defined as time from HCC diagnosis to death. Criteria for censoring included loss to follow-up or end of study follow-up (1 December 2019) whichever was earlier.

### 2.3. Statistical Analysis

Continuous variables were expressed as median (range), and the Mann–Whitney U test was used to compare continuous variables. Numbers and percentages were used to describe the distribution of categorical variables. Pearson Chi-Squared and Fisher’s exact tests were used to compare categorical variables. We used the Kaplan–Meier method to evaluate patient survival and the log-rank test to compare survival statistics among the study subgroups.

We performed the following subgroup analyses: viremic HCC vs. post-SVR HCC, and viremic HCC with subsequent SVR, viremic HCC who remained viremic vs. post-SVR HCC in BCLC stage 0/A, B, and C/D patients.

We used univariable and multivariable Cox’s proportional hazard analysis to evaluate factors associated with overall survival. The criteria used to select for variable included in the multivariable model were by *p* value < 0.05 in univariate analysis.

In addition, we performed a sensitivity analysis comparing overall survival in patients who developed HCC >1 year after HCV SVR to avoid the confounding of pre-existing/prevalent HCC in post-SVR HCC patients. We also performed propensity score matching (PSM) on variables showing significance in the survival analysis (detail variables described in results) to balance the background risk between the post-SVR HCC and the viremic HCC groups (PSM analysis 1), and between the post-SVR HCC and the viremic HCC with subsequent SVR (PSM analysis 2). We then compared the survival outcomes between the groups from the two PSM analyses. In multivariable regression analysis, to avoid further decrease in sample size, missing data were managed by imputation using the sample median value for continuous variables and coding “other/missing” for categorical variables [16].

All tests were two-sided, and *p* < 0.05 was considered significant. All analyses were performed using the SPSS 20.0 statistical package (SPSS Inc., Chicago, IL, USA).

## 3. Results

### 3.1. Study Population and Patient Characteristics at HCC Diagnosis

A total of 1389 HCV- HCC patients, including 301 post-SVR HCC and 1088 viremic HCC, were enrolled (Figure 1). Compared to viremic HCC, post-SVR HCC were older, less likely male, less likely obese, and less likely to have a history of alcohol use, smoking, hypertension, or metabolic disorders. They were also less likely to have fibrosis-4 (FIB-4) index > 3.25, liver cirrhosis or CPT ≥ 7. They also had lower AFP level. (Table 1) Among post-SVR HCC patients, HCC developed at a median of 1.7 years after antiviral HCV eradication. Among viremic HCC patients, 239 out of the total of 1088 patients were subsequently treated and achieved SVR at a median of 1.1 years after HCC diagnosis (viremic HCC with subsequent SVR).

### 3.2. Tumor Characteristics at HCC Diagnosis and Initial Treatment Modalities

Compared to viremic HCC patients, post-SVR HCC patients had lower BCLC stage, higher proportion of solitary tumor, smaller tumor size, and less extra-hepatic metastasis. (Table 2) Patients of post-SVR HCC had a higher rate of initial primary HCC treatment with surgery, but not other curative therapies. The proportion of initial primary treatment with palliative therapy was also lower in post-SVR HCC patients.

Among the 143 patients who underwent partial hepatic resection, we observed no significant difference in tumor characteristics (e.g., cellular differentiation, pathological stage, and micro-vascular invasion, or non-tumor part fibrosis stage) between the post-SVR HCC and the viremic HCC groups. (Appendix A) Notably, 40% of the patients in the post-SVR HCC group and 31.7% of the patients in the viremic HCC groups had fibrosis stage of F0-2 in histological examination of the non-tumorous liver tissues.

### 3.3. Overall Survival between post-SVR HCC and Viremic HCC 

The median overall survival was 61.5 months (95% CI: 53.2–69.7) months for the entire study cohort, with the post-SVR HCC group having significantly longer survival compared to the viremic HCC group (153.3 vs. 55.6 months, *p* < 0.01). (Figure 2A) However, on sub analysis by BCLC stage, the survival difference between the two study groups was only found among BCLC stage 0/A patients, but not among those with BCLC stage B or C/D (Figure 2B–D).

On univariable Cox regression analysis, post-SVR HCC (vs. viremic HCC) were associated with better survival as well as BCLC stage 0/A (vs. B/C/D), curative therapy (vs. palliative therapy/supportive care). (Table 3) On multivariable analysis, there was no longer significant difference between post-SVR HCC vs. viremic HCC in association to survival after adjustment for potential confounders, while the presence of liver cirrhosis, higher AST, ALT, creatinine, AFP, BCLC stage 0/A, and curative therapy were independently associated with overall survival.

On sub analysis of the viremic HCC group, we found significant differences in survival rates among the groups with the highest median survival in the viremic HCC with subsequent SVR group, followed by post-SVR HCC patients, and the lowest among those who remained viremic (*p* < 0.01). (Figure 3A) In addition, the survival differences remained significant in subgroup analysis by the BCLC stage with the viremic HCC with the subsequent SVR group having the highest survival (*p* < 0.01 for all BCLC stages). (Figure 3B–D) On multivariable Cox regression analysis, viremic HCC with subsequent SVR was an independent factor associated with lower overall mortality compared to post-SVR HCC (adjusted HR: 0.18, 95% CI: 0.11–0.29, *p* < 0.01), and there was no significant difference between viremic patients who remained viremic and the post-SVR group (adjusted HR: 1.41. 95% CI: 1.00–1.98, *p* = 0.05) (Table 4).

To avoid the confounding of potential pre-existing HCC in post-SVR HCC patients, we performed a sensitivity analysis by excluding 68 patients who developed HCC within one year after SVR and found similar results with viremic HCC with subsequent SVR having significantly lower mortality than post-SVR HCC (adjusted HR: 0.16, 95% CI: 0.09–0.27, *p* <0.01). (Appendix A) We further performed the propensity score matching (PSM) on variables showing significance in our survival analysis to balance the two comparison groups. The PSM (on liver cirrhosis, impaired liver status, AST, ALT, creatinine, AFP, BCLC stage 0/A, and curative therapy) to compare post-SVR and viremic HCC patients yielded 140 pairs of post-SVR HCC and viremic HCC patients who were comparable in most characteristics. (Appendix A). We found no significant survival difference between the two groups, (Appendix A) and there was no significance between the two groups in their association to survival on multivariable Cox regression analysis (adjusted HR: 0.93, 95% CI: 0.60–1.45, *p* = 0.76). (Appendix A) However, viremic HCC with subsequent SVR was again associated with improved survival compared to post-SVR HCC (adjusted HR: 0.19, 95% CI: 0.04–0.81, *p* = 0.03). (Appendix A) Lastly, we performed a second PSM to match the viremic HCC with subsequent SVR and the post-SVR HCC patients by liver cirrhosis, impaired liver status, AST, ALT, creatinine, AFP, BCLC stage 0/A, curative therapy, and HCC diagnosis year yielding 80 pairs of patients with comparable characteristics. (Appendix A) In this analysis, we also found a higher overall survival in viremic HCC with subsequent SVR patients compared to post-SVR HCC patients, (Appendix A) and that viremic HCC with subsequent SVR was independently associated with better survival (adjust HR: 0.15, 95% CI: 0.05–0.39, *p* < 0.01). (Appendix A) Thus, the results from the two PSM analyses were consistent with the results observed in the analysis of the total cohort.

## 4. Discussion

Our large multinational study found that viral status at the time of HCC diagnosis is an important factor affecting patient presentation as well as survival outcomes. Patients who developed HCC after SVR had better liver function, significantly lower tumor stage compared to patients who were viremic at the time of HCC diagnosis (BCLC 0/A: 74.8% vs. 46.1%), as well as higher median survival among those with BCLC 0/A stage. Interestingly, we found that patients who were viremic at HCC diagnosis but subsequently received antiviral therapy and achieved SVR had even better survival than those who developed HCC post-SVR; and as expected, the viremic patients who remained viremic had the lowest survival. This finding expands on prior studies that found higher survival in patients who received DAA therapy and achieved SVR after HCC diagnosis compared to untreated HCV patients [11,17].

Our finding of older patients, better liver function and lower proportion of patients with cirrhosis among the post-SVR HCC group is in line with prior studies [18]. However, while our post-SVR HCC cohort had lower tumor stage, tumor size, and similar tumor differentiation grade, others have reported either similar tumor stage or larger tumor size, more advanced tumor stage and/or less favorable tumor differentiation grade [18,19]. The discrepancies between studies may be related to the differences among study cohorts due to different ethnicity/genetic background [20] and/or different post-SVR HCC surveillance strategy [6]. Currently, the professional society guideline generally recommends HCC surveillance post-SVR only in patients with cirrhosis [14], but our study found that one in three patients who developed HCC after achieving SVR did not have cirrhosis; and among the subgroups with histologic examination of resected liver tissues, 55% of those who developed HCC after SVR did not have cirrhosis and about 40% only had stage 0–2, consistent with prior reports of HCC development in HCV patients without cirrhosis [21]. Regardless, HCC surveillance after HCV eradication is critical and further studies are needed to identify high risk groups who require continued long-term surveillance following SVR.

As previously noted, prior studies focused on the effect of anti-HCV therapy after HCC developed and have shown improved liver-related and overall survival [11,17,22,23,24], but little is known about the long-term outcome among patients with post-SVR HCC. In the current study, we observed that patients with post-SVR HCC had better survival than the patients who developed HCC while having HCV viremia. However, post-SVR HCC patients lost the advantage in survival after adjustment for the other potential confounders such as liver function and tumor staging, suggesting that the survival advantage was largely due to better liver function due to HCV eradication and lower tumor staging that could be due to better surveillance in this population. Another potential explanation was that those who developed HCC after viral eradication may have other host genetic risks predisposing them to poor outcomes that were not measurable and adjusted for in our studies. Prior studies have reported that the HCC risk gene signature was not reversed in those who developed HCC after response to IFN-based therapy [20], and HCV-induced epigenetic “scar” associated with hepatocarcinogenesis persists after viral eradication [25], though whether the “left-over” epigenetic scar predisposing patients to HCC development can also affect survival outcome remains to be studied.

HCC curative therapies included liver transplantation, ablation, and resection [26]. Although, there were non-significant differences in the survival among these three different curative therapies, the long-term outcome was substantial better in the patients receiving liver transplantation or resection than in those receiving ablation. In one recent paper, the 5-year overall survival was 70% and 60% in HCC patients who received live resection and ablation (*p* = 0.666) [27]. A same high survival and low recurrence was also observed in patients who received liver transplant [28]. Because of the non-significant difference of survival and small case numbers of liver transplant, we grouped, instead of separated, patients of these three difference curative therapies to compare with palliative/supportive therapies.

We acknowledge the following limitations. First, this was a retrospective study and subject to inherent limitations of retrospectively collected data, but we treated the missing data as described in the methods and performed additional analysis using propensity score matching to balance background risks of comparative groups to minimize the effect of confounders. Second, we did not have data on potential additional confounding factors such as indication or criteria for HCV therapy which can introduce bias that can affect disease monitoring and surveillance that can affect survival, beyond the demographic, clinical and treatment characteristics we identified. Thus, a prospective well controlled study is needed to confirm our findings.

## 5. Conclusions

In conclusion, we demonstrated that favorable clinical and tumor characteristics contributed to better survival among patients with post-SVR HCC compared to patients who developed HCC while being viremic overall, and viremic HCC patients had lower survival than post-SVR patients and viremic patients who were treated and achieved SVR after HCC diagnosis. Together, our data support timely antiviral treatment for HCV patients both before and after HCC diagnosis. In addition, as 30% of post-SVR HCC patients did not have known cirrhosis at the time of HCC occurrence, HCC surveillance should not be restricted to only post-SVR patients with cirrhosis, and further studies are needed to develop a cost-effective strategy to identify post-SVR patients without known cirrhosis who remain at high risk for HCC surveillance.

## Figures and Tables

**Figure 1 cancers-13-03455-f001:**
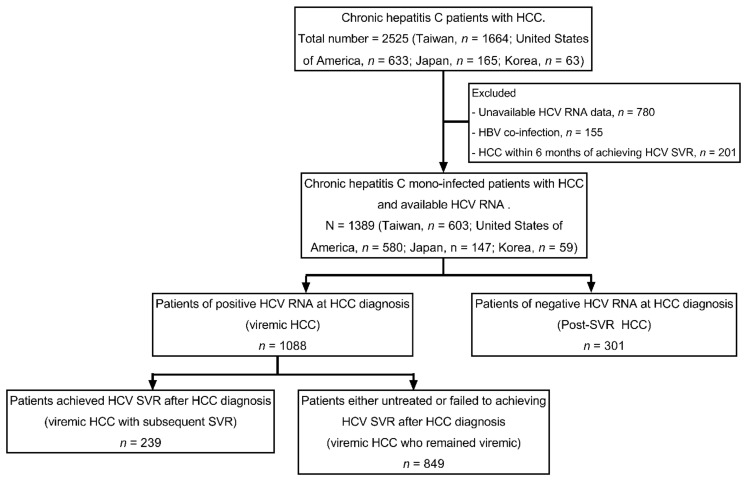
Study allocation flow chart. HCC, hepatocellular carcinoma; HCV, hepatitis C virus; HBV, hepatitis B virus; SVR, sustained virological response.

**Figure 2 cancers-13-03455-f002:**
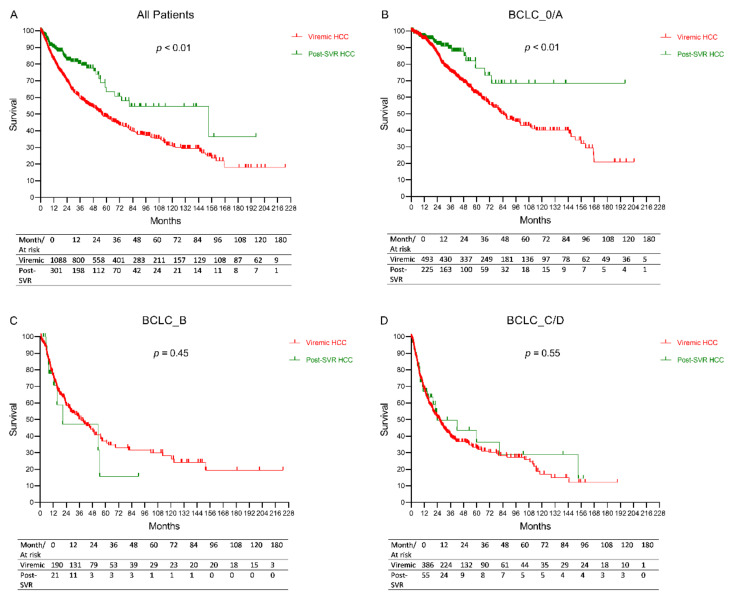
Overall survival of post-SVR HCC versus viremic HCC in the total cohort and by BCLC stage. (**A**) All patients, (**B**) BCLC stage 0/A, (**C**) BCLC stage B, (**D**) BCLC stage C/D.

**Figure 3 cancers-13-03455-f003:**
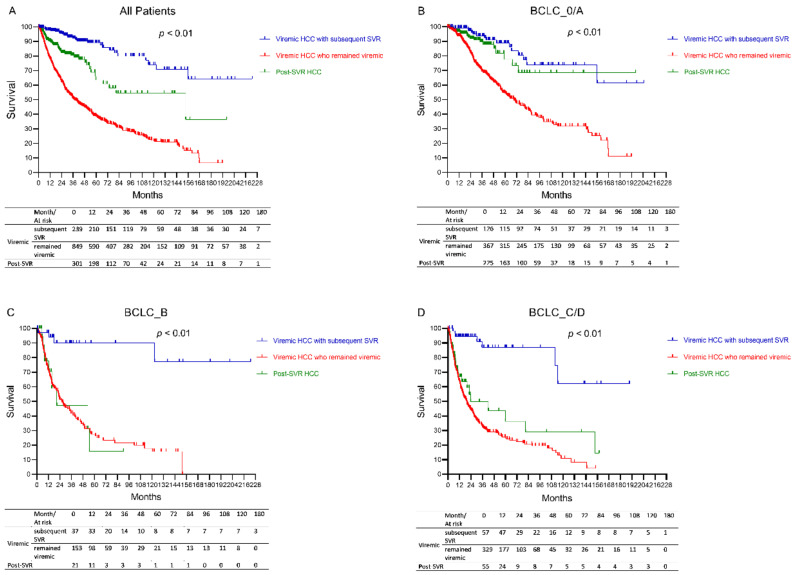
Overall survival in post-SVR HCC, viremic HCC with subsequent SVR and viremic HCC who remained viremic in the total cohort and by BCLC stage. (**A**) All patients, (**B**) BCLC stage 0/A, (**C**) BCLC stage B, (**D**) BCLC stage C/D.

**Table 1 cancers-13-03455-t001:** Baseline characteristics and clinical features at HCC diagnosis.

Patient Characteristics	Viremic HCC *n* = 1088	Post-SVR HCC *n* = 301	*p*
Age, years	62 (30–89)	70 (21–94)	<0.01
Male sex	711 (65.3)	166 (55.1)	<0.01
BMI, kg/m^2^	25.0 (14.6–52.5)	24.2 (14.7–51.1)	<0.01
Obesity	205/757 (27.1)	55/267 (20.6)	0.04
Alcohol use	374/1032 (36.2)	45/288 (15.6)	<0.01
Smoking	428/1031 (41.5)	44/288 (15.3)	<0.01
Hypertension	476/1032 (46.1)	88/289 (30.4)	<0.01
Diabetes	321/1063 (30.2)	81 (26.9)	0.28
Metabolic disorders	680/1048 (64.9)	141/292 (48.3)	<0.01
Liver cirrhosis	850 (78.1)	205 (68.1)	<0.01
Impaired liver status	367/1072 (34.2)	26/144 (18.1)	<0.01
Platelet, × 10^3^/uL	116 (5–1374)	144 (33–348)	<0.01
Albumin, g/dL	3.6 (1.2–6.7)	4.1 (2.1–4.9)	<0.01
Total bilirubin, mg/dL	1.0 (0.2–25.6)	0.9 (0.2–27.7)	<0.01
Prothrombin time INR	1.10 (0.80–4.30)	1.08 (0.91–2.10)	<0.01
AST, U/L	77 (11–3008)	33 (17–812)	<0.01
ALT, U/L	67 (6–1949)	28 (9–587)	<0.01
Creatinine, mg/dL	0.9 (0.4–14.4)	0.9 (0.3–7.6)	0.71
AFP, log_10_ ng/mL	1.4 (0.2–5.6)	1.0 (−0.01–4.7)	<0.01
FIB-4 index >3.25	784/1060 (74.0)	70/141 (49.6)	<0.01
HCV SVR to HCC, year	-	1.7 (0.5–13.5)	-
HCC to HCV SVR, year ^†^	1.1 (0.2–15.7)	-	-

Continuous variables were presented with median (range), and statistics with Mann–Whitney U test. Categorical variables were presented with numbers (percentage), and statistics with Chi-Square and Fisher’s exact test. Metabolic disorder: Obesity, hypertension, or diabetes. Impaired liver status: Child-Pugh score ≥7. BMI, body mass index; INR, international ratio; AST, aspartate aminotransferase, ALT, alanine aminotransferase; AFP, alpha-fetoprotein. ^†^ 239 viremic HCC patients subsequently received antiviral therapy and achieved SVR after HCC diagnosis (viremic HCC with subsequent SVR).

**Table 2 cancers-13-03455-t002:** HCC characteristics and treatment modalities.

Tumor Characteristics	Viremic HCC *n* = 1088	Post-SVR HCC *n* = 301	*p*
Clinical characters	-	-	-
BCLC stage	-	-	<0.01
0	128 (12.0)	162 (53.8)	-
A	365 (34.1)	63 (20.9)	-
B	190 (17.8)	21 (7.0)	-
C	322 (30.1)	50 (16.6)	-
D	64 (6.0)	5 (1.7)	-
BCLC stage 0/A	493/1069 (46.1)	225 (74.8)	<0.01
Solitary tumor	641/1056 (60.7)	239/297 (80.5)	<0.01
Largest tumor size, cm	2.6 (0.5–18.7)	2.0 (0.7–20.5)	<0.01
Largest tumor ≥5 cm	205/1056 (19.4)	20/157 (12.7)	0.05
Macroscopic vessel invasion	84/1077 (7.8)	20/300 (6.7)	0.62
Extra-hepatic metastasis	70/949 (7.4)	7/298 (2.3)	<0.01
Primary HCC treatment	-	-	-
Curative therapy	445/1047 (42.5)	81/160 (50.6)	0.06
Surgery	212/1047 (20.2)	45/160 (28.1)	0.03
Local ablation	205/1047 (19.6)	34/160 (21.3)	0.60
Liver transplant	28/1047 (2.7)	2/160 (1.3)	0.41
Palliative therapy	477/1047 (45.6)	59/160 (36.9)	0.04
Supportive care	121/1061 (11.4)	16/164 (9.8)	0.60

Continuous variables were presented with median (range), and statistics with Mann– Whitney U test. Categorical variables were presented with numbers (percentage), and statistics with Chi-Square and Fisher’s exact test, and Pearson Chi-Square test. BCLC, Barcelona clinic liver cancer; LT, liver transplantation; RFA, radiofrequency ablation; PEI, pure ethanol injection; TACE, transhepatic arterial chemoembolization. Local ablation therapy includes radiofrequency ablation and pure ethanol injection therapy.

**Table 3 cancers-13-03455-t003:** Factors associated with overall mortality in viremic vs. post-SVR HCC.

	Univariate		Multivariate	
	HR (95% CI)	*p*	HR (95% CI)	*p*
Age, years	1.00 (0.99–1.00)	0.25	-	-
Male sex	1.27 (1.06–1.51)	0.01	1.13 (0.92–1.38)	0.24
Obesity	1.33 (1.07–1.67)	0.01	1.16 (0.92–1.46)	0.20
Alcohol use	1.52 (1.28–1.81)	<0.01	1.02 (0.82–1.27)	0.89
Smoking	1.31 (1.11–1.56)	<0.01	0.89 (0.73–1.10)	0.29
Metabolic disorders	1.17 (0.98–1.40)	0.08	-	-
Liver cirrhosis	2.44 (1.92–3.12)	<0.01	1.81 (1.40–2.35)	<0.01
Impaired liver status	2.19 (1.85–2.60)	<0.01	1.19 (0.97–1.48)	0.10
Platelet, × 10^3^/uL	1.00 (1.00–1.00)	0.10	-	-
AST, U/L	1.00 (1.00–1.00)	<0.01	1.00 (1.00–1.00)	<0.01
ALT, U/L	1.00 (1.00–1.00)	<0.01	1.00 (1.00–1.00)	0.02
Creatinine, mg/dL	1.11 (1.04–1.18)	<0.01	1.14 (1.07–1.22)	<0.01
AFP, log_10_ ng/mL	1.74 (1.61–1.89)	<0.01	1.55 (1.43–1.69)	<0.01
BCLC 0/A (vs. B/C/D)	0.35 (0.30–0.42)	<0.01	0.54 (0.44–0.66)	<0.01
Curative therapy (vs. palliative therapy/supportive care)	0.47 (0.39–0.56)	<0.01	0.60 (0.49–0.73)	<0.01
Viremic HCC	1	-	1	-
Post-SVR HCC	0.51 (0.39–0.68)	<0.01	1.05 (0.76–1.47)	0.76

Metabolic disorder: Obesity, hypertension, or diabetes. Impaired liver status: Child-Pugh score ≥7.

**Table 4 cancers-13-03455-t004:** Factors associated with mortality in post-SVR and subgroups of viremic HCC patients.

Variables	Univariate		Multivariate	
	HR (95% CI)	*p*	HR (95% CI)	*p*
Age, years	1.00 (0.99–1.00)	0.25	-	-
Male gender	1.27 (1.06–1.51)	0.01	1.12 (0.92–1.37)	0.27
Obesity	1.33 (1.07–1.67)	0.01	1.20 (0.95–1.51)	0.13
Alcohol	1.52 (1.28–1.81)	<0.01	1.12 (0.90–1.39)	0.31
Smoking	1.31 (1.11–1.56)	<0.01	0.93 (0.76–1.14)	0.50
Metabolic disorders	1.17 (0.98–1.40)	0.08	-	-
Liver cirrhosis	2.44 (1.92–3.12)	<0.01	1.92 (1.48–2.49)	<0.01
Impaired liver status	2.19 (1.85–2.60)	<0.01	1.26 (1.02–1.56)	0.03
Platelet, × 10^3^/uL	1.00 (1.00–1.00)	0.10	-	-
AST, U/L	1.00 (1.00–1.00)	<0.01	1.00 (1.00–1.00)	<0.01
ALT, U/L	1.00 (1.00–1.00)	<0.01	1.00 (1.00–1.00)	0.02
Creatinine, mg/dL	1.11 (1.04–1.18)	<0.01	1.16 (1.09–1.24)	<0.01
AFP, log_10_ ng/mL	1.74 (1.61–1.89)	<0.01	1.52 (1.40–1.66)	<0.01
BCLC 0/A (vs. B/C/D)	0.35 (0.30–0.42)	<0.01	0.57 (0.46–0.69)	<0.01
Curative therapy (vs. palliative therapy/supportive care)	0.47 (0.39–0.56)	<0.01	0.60 (0.49–0.73)	<0.01
Post-SVR HCC	1	-	1	-
Viremic HCC who remained viremic	2.52 (1.90–3.35)	<0.01	1.41 (1.00–1.98)	0.05
Viremic HCC with subsequent SVR	0.37 (0.23–0.59)	<0.01	0.18 (0.11–0.29)	<0.01

Metabolic disorder: Obesity, hypertension, or diabetes. Impaired liver status: Child-Pugh score ≥7.

## Data Availability

The data presented in this study are available within the article and Appendix A.

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
