# Peer review of "Characteristics and Survival Outcomes of Hepatocellular Carcinoma Developed after HCV SVR"

_cancers, 2021, doi:10.3390/cancers13143455_

Round 1
Reviewer 1 Report
Yeh et al. submitted a manuscript entitled “Characteristics and Survival Outcomes of Hepatocellular Carcinoma Developed After HCV SVR”. The authors described a retrospective study of a multinational cohort of 1,389 HCC patients who were viremic for HCV (1,088) or post-SVR (301) at the time of HCC diagnosis. The authors compared the two groups of patients but also post-SVR group with a group of patients (289) who subsequently after HCC diagnosis received anti-viral treatment and achieved SVR. Please, find below some questions that can help improve the manuscript.
- Figure 1: what is the number of patients for each country among the 1,389 ?
- Line 170: could the number 143 be introduced line 106 ?
- Figure 2 and 3: could the authors check the readability of tables ?
- Lines 221-222: could the authors clarify the 6 months/1 year criteria from the two sentences “excluding 68 patients who developed HCC within one year after SVR” and “We excluded patients with … HCC diagnosis within 6 months of achieving HCV SVR” (line 79- 80) ?
- Could the authors hypothesize mechanisms explaining better survival of patients receiving antiviral after HCC diagnosis and achieving SVR in comparison of patient with HCC post SVR ?
Reviewer 2 Report
In their paper entitled Characteristics and Survival Outcomes of Hepatocellular Carcinoma Developed After HCV SVR Ming-Lun Yeh and colleagues explored the oncological outcome of patients diagnosed with HCC according to viral replication state and SSVR after DAA/INF treatment
I find several limitation to this paper that should be adressed:
1) Why to exclude those patients who were diagnosed with HCC between 6 months from SVR? Specify this in the methods
2) Considering that the reported results could be strongly affected by selection biases, I think that OS should not be regarded as a reliable "oncological" outcome measure - Authors should implement their paper through a competing risk analysis for tumor-related death in order to provide a correct interpretation of their data
3) Although LT, ablation and resection represent curative strategies, their collective inclusion in the same treatment group is not proper as those treatments obviously results in different survival (and tumor-related survival) outcomes
4) ECOG, WBC and Hb abbreviations are reported table 1 and 2, footnotes but not presented in any of the two tables, while metabolic disorder is repeated
4) I noticed that the BMI ranges until a value of 51 - is that correct??
5) Looking at the total number of patients for each variable reported in Table 2, I identified several variables with important missings that are note explicitly highlighted in the text (i.e. primary HCC treatment are reported for 160 out of the 301 patients); these important loss of data severely impairs the study results
6) Risk factors with a HR of 1 and a confidence interval of (1-1) could not be statistically significant
I am afraid that the paper should undergo strong statistical review before reconsideration
Round 2
Reviewer 2 Report
The Authors partially addressed several points
The choice to group liver transplantation, ablation and resection in the same category of “curative treatments” should be furtherly discussed: I suggest the Authors to refer to these recently published papers while focusing on this aspect:
1) LT - doi: 10.3390/cancers11060741
2) Resection vs. Ablation - doi: 10.1016/j.ejso.2019.04.023
Best regards
Round 3
Reviewer 2 Report
Authors’ revision significantly improved manuscipt quality
There is some typo error at line 79 with the reference number: (14 , 15) 3) - did you mean (3, 14, 15)?
First Author’s surname of for reference 27 and 28 is wrong: the actual surname is DI SANDRO, S.
Please, provide the DOI for each reference following the instruction for Authors (https://www.mdpi.com/authors/references)
Congratulations for your effort
Best regards
